# Information advantage in sensing revealed by Fano-resonant Fourier scatterometry

Nick Feldman[1,2], Arie J. den Boef[2,3,4], Lyubov V. Amitonova[2,3] & A. Femius Koenderink [1] ✉

Fano resonances in nanophotonic structures are attractive for sensing due to their ultanarrow resonant linewidths and high local fields. Conventional read out schemes rely on measuring a frequency shift in Fano scattering spectra as function of perturbation. We experimentally demonstrate that angle-resolved analysis of the scattering of a Fano resonant structure is quantitatively more informative than measuring spectral shifts. We theoretically discuss how a perturbation affects fundamental nanophotonic properties of a Fano resonant metasystem, and how these are transduced to an observable far field response. We perform a rigorous experimental study in which we characterize deeply subwavelength perturbations in a Fano resonant dielectric metasurface using a conventional spectral approach, and a Fourier scatterometry based approach, and show that perturbations can lead to marked directional scattering in Fourier space. We finally quantitatively compare these two sensing methods in terms of their inherent Fisher information content, and show that an information advantage is obtained when the signal is resolved in Fourier space.

Resonance based sensors are at the heart of many modern day scientific and commercial instruments to accurately detect minute forces[1] or mass changes[2], characterize material properties[3] and acquire high quality medical images[4]. In these technologies, the strong responsivity of the system at resonance is exploited to retrieve precise estimations of the relevant physical property to be detected. In the field of nano-optics, resonance based sensors are routinely exploited to detect perturbations in the form of biomolecules[5,6], gases[7,8], displacements[9] and accelerations[10], where the resonant response originates from localized surface plasmon modes in noble metal nanoparticles[5,6] or an interplay between optical and mechanical vibrations[9,10], to name a few. In all these nanophotonic sensing protocols, a resonant spectral feature is usually identified in experimental observables like absorption[5], scattering and extinction cross sections[6,7] in single nanoparticles, or reflection and transmission spectra in extended thin films or arrays of nanoparticles[6]. Refractive index changes and geometry changes form a perturbation that imprints a detectable shift onto the resonance frequency. To achieve optimal sensing performance, which is usually captured in the sensing figure of merit (FOM), the shift of the resonance frequency per unit perturbation needs to be large compared to the linewidth of the resonance, and systems which exhibit ultrahigh quality factor (Q-factor) resonant features are therefore highly sought after in the sensing community. The Fano resonance[11,12], which is a distinct asymmetric spectral line shape arising from interference effects between a broad superradiant and sharp subradiant state, has shown great potential in sensing applications due to their inherent high Q-factors[13]. The recent advent of dielectric metasurfaces, which are two-dimensional (2D) sheets decorated by designer low-loss dielectric particles, has provided a powerful toolbox for engineering these Fano resonances through the physics of bound states in the continuum (BIC)[14,15]. A BIC is a phenomenon in which a state remains perfectly localized and therefore does not radiate, despite existing within the continuous energy spectrum of radiating states. True BICs therefore have a vanishing resonance linewidth and infinite quality factor, and can only exist within infinite and ideal lossless systems. In practical scenarios a BIC mode can be translated into a so-called quasi-

[1]Department of Information in Matter and Center for Nanophotonics, AMOLF, Amsterdam, The Netherlands. [2]Metrology department, Advanced Research Center for Nanolithography (ARCNL), Amsterdam, The Netherlands. [3]Department of Physics and Astronomy, and LaserLaB, Vrije Universiteit, Amsterdam, The Netherlands. [4]ASML Netherlands B.V., Veldhoven, The Netherlands. ✉e-mail: f.koenderink@amolf.nl

BIC with finite and tunable quality factor by introducing a small asymmetry in the structural parameters of the metasystem. Sensors exploiting the excitation of these symmetry protected quasi-BIC modes have shown record high sensing FOM's[16,17].

In parallel to these resonance based efforts, nanophotonic platforms have been designed that efficiently transduce information about a given perturbation into the far-field by monitoring the rich contents of angle resolved scattering patterns. Indeed, deeply subwavelength or "superresolution" estimation precision of a perturbation has been achieved by analyzing the Fourier space scattering of complex nanophotonic objects[18,19], and even single nanoparticles[20–22]. These studies differ in experimental observable from the resonance based works in the sense that in the former, the far-field signal is resolved into its angular components (Fourier space), whereas in the latter the signal is usually acquired in an angle integrated modality on either a spectrometer or a single pixel bucket detector, thereby losing potentially useful information about the perturbation in Fourier space. In fact, Fano resonances have been shown to exhibit directionality properties[23,24], which could provide an information gain in a sensing based experiment. However, to the best of our knowledge, there are no reported sensing experiments in which the signal of a Fano resonant scattering sensor is analyzed in Fourier space, and in which the information advantage is quantified which such an angle resolved measurement modality may have relative to conventional spectroscopy.

In this work, we study how perturbations affect the far field scattering properties of a Fano resonant metasystem, and show that an information gain can be achieved when the resonant scattering is analyzed in Fourier space, as conceptually sketched in Fig. 1. We start with a theoretical discussion of all relevant nanophotonic response function changes a perturbation can induce onto a Fano resonant system, and how these are transduced into the far-field. We then perform a rigorous experimental study, in which we characterize the influence of a perturbation in dielectric Fano resonant metasystems in both an angle integrated spectral modality and a Fourier space measurement, and show that deeply subwavelength information about the perturbation is resonantly transduced into marked directional scattering in Fourier space. We finally compare both measurement modalities in terms of the Fisher information they contain about the perturbation, and show quantitatively that a Fourier space measurement has a higher Fisher information content than an angle integrated measurement.

## Results

### Theoretical motives

To study how a Fano resonant metasystem transduces information into far-field Fourier components, our first objective is to identify a Fano resonant metasurface that allows light scattering into higher order diffractive channels. This is not a trivial prerequisite, since the majority of Fano resonant metasurfaces studied in the literature are constructed by semi-infinite periodic lattices with subwavelength periodicity, thus only allowing coupling to the zero order transmission and reflection channels. Possible disturbances from the perfect subwavelength periodicity, that are necessary to generate diffraction channels, such as edge effects[25,26] and fabrication imperfections[27] generically lead to a reduction of the resonance quality factor. Besides semi-infinite non-diffractive lattices, other well known types of metasystems supporting Fano resonances are for instance plasmonic[28,29] and dielectric[30] oligomers of nanoparticles, and even single dielectric nanoparticles supporting so-called supercavity modes[30–32]. In contrast to the sub-diffractive infinitely extended metalattices, these finite sized systems are allowed to scatter light into a continuous angular spectrum, fully determined by the structure factor of the relevant mode profile at play. Drawbacks of many finite-sized Fano resonant structures are that often the quality factor is not easily tunable, or that complex structured illumination conditions are required to selectively

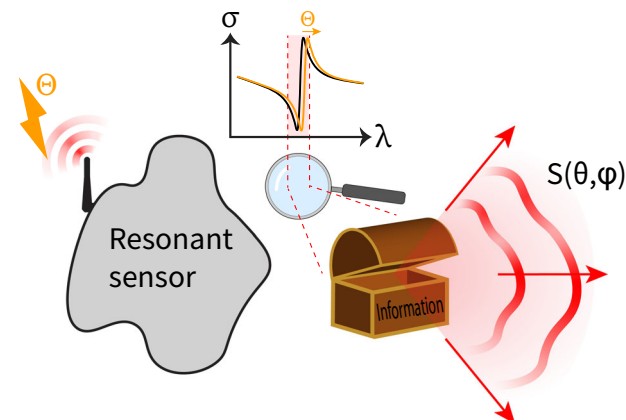

**Fig. 1 | A Fano resonant scattering sensor detects a perturbation $\Theta$, causing the resonance wavelength to be shifted.** When the signal is analyzed in Fourier space, however, additional information about the perturbation can be gained.

excite the resonances. We identify symmetry protected quasi-BIC Fano resonances in finite sized annular metarings as particularly promising for our study. Such metarings were demonstrated in a recent study by Kühner et al.[33] and have tuneable resonances that are accessible by simple plane wave illumination. While the interparticle spacing of the high refractive index meta-atoms in such rings is still subdiffractive to ensure strong collective mode coupling, the shape in which the meta-atoms are patterned allows light scattering into higher order Fourier channels. In the language of X-ray scattering or treatments of antenna directivity, the radiation pattern can be understood as the product of a 'structure factor', or 'array factor', that derives from the placement of units on a circle, and a 'form factor' or 'element factor' that accounts for the dipolar radiation pattern of each individual meta-atom[34]. The structure factor of a circle, chosen as a few wavelengths across, provides concentric rings of scattered intensity in Fourier space. The flexible toolkit of symmetry protected quasi-BIC resonances allows relatively straightforward tuning of the Fano resonance by controlling structural symmetry parameters.

In the following, we will focus on sensing perturbations in the form of deeply subwavelength structural displacements between the individual building blocks in meta rings as described above. This type of perturbation is frequently encountered in modern-day semiconductor chip manufacturing processes, where nanometer-scale misalignments between individual structures on a device layer can arise when the same wafer goes through several successive exposure/etch/exposure cycles where each exposure step only defines part of the structure. Accurate monitoring of these misalignments, which is termed "interlaced metrology"[35,36], is crucial to ensure high-quality chip performance. This is typically achieved by analyzing the diffracted signal of devoted scattering sensors that are printed during the same exposure as the actual devices in the chip. Currently, there is a large push within the semiconductor industry for small footprint scattering sensors containing an effective footprint of no more than $2 \times 2\,\mu m$. Since the meta rings contain a diameter of only $3\,\mu m$, which could be reduced even further if needed, we envision that these structures could be exploited as responsive, yet small footprint nanophotonic scattering sensors for diffraction based interlaced metrology applications. Figure 2a, b illustrate the scenario at hand on basis of electron microscopy images of resonant silicon metarings that we interrogate in the experimental part of this paper (see methods section for nanofabrication details). Figure 2a displays an unperturbed metaring: radially oriented silicon blocks are equidistantly placed on a ring, and blocks alternate in length. The asymmetry in the length of neighboring meta-atoms $\Delta L$ tunes the Fano resonance by a symmetry protected quasi-BIC mechanism[15,33]. In Fig. 2b, the perturbed structure is shown.

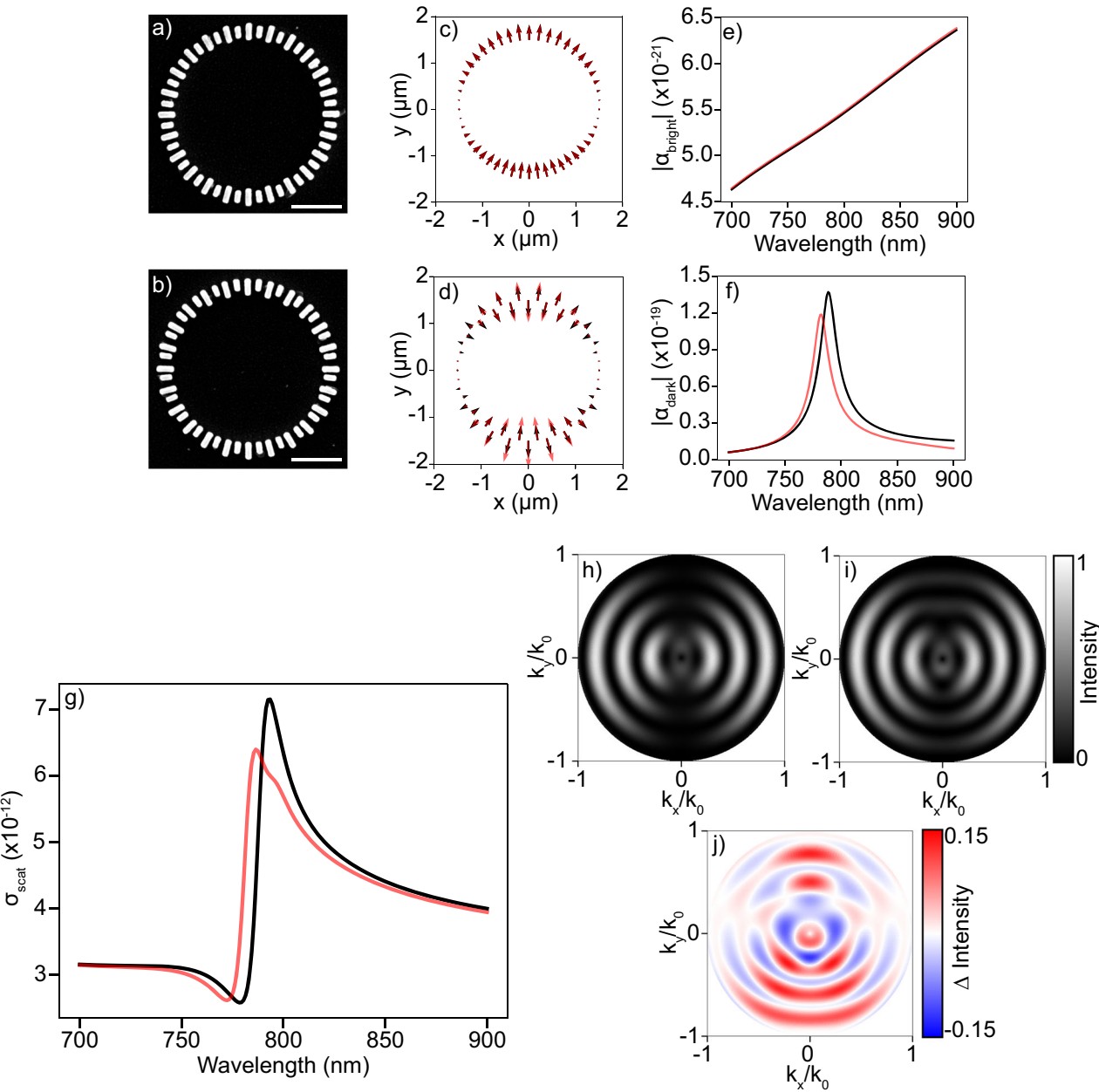

**Fig. 2 | Theoretical considerations in Fano resonance based sensing schemes.** Panels **a** and **b**: SEM images of the unperturbed and perturbed versions of the studied Fano resonant metasystem, respectively, where the perturbation corresponds to an upward shift of every other meta-atom by 50 nm. Scalebar corresponds to 1 μm. Panels **c** and **d**: respectively super and subradiant eigenmode profiles of the dipole orientations in the metaring. The black and red arrows showcase unperturbed and perturbed versions of the eigenmodes, respectively. Panels (**e**) and (**f**): Absolute value of the modal excitation coefficient for the super and subradiant eigenmodes, respectively. Panel **g**: Scattering cross section of the (un)perturbed dipolar metasystem. Panels **h** and **i**: Fourier space radiation patterns for the unperturbed and perturbed metaring, respectively, driven by an excitation wavelength of 775 nm. Panel **j**: Differential radiation pattern resulting from the subtraction of radiation patterns in panels (H) and (I).

The perturbation corresponds to a collective upward shift $\Delta y$ of the entire subring of short meta-atoms.

To explore the influence of perturbations in a Fano resonant sensing scenario, we tackle the problem from an eigenmode perspective. While one could derive quasinormal modes from a numerical eigensolver, we semi-analytically model the annular metasystem using coupled dipole theory[37,38], in which the individual meta-atoms are approximated as polarizable point dipoles, while all electrodynamic multiple scattering interactions are taken into account. Using the specific modeling parameters as discussed in the methods section, we construct the dipole-dipole interaction matrix

of the system, from which we extract all relevant modes of the metarings by performing an eigenvalue decomposition. We obtain (essentially frequency-independent) eigenvectors $\mathbf{p}_m$, which correspond to the natural coupled dipole oscillations in the metaring, along with corresponding eigenvalues $\lambda_m(\omega)$, describing the eigenpolarizabilities of the respective modes. The response of the metaring to incident light can be decomposed in the eigenmodes, and depends both on the magnitude of the eigenpolarizabilities and the overlap of the incident field with the eigenvector. We assume an externally applied electric field $\mathbf{E}_{exc}$ that correspond to a y-polarized plane wave normally incident onto the ring. This will lead to a driven

dipole configuration $\mathbf{p}_{\text{driven}}$

$$\mathbf{p}_{\text{driven}} = \sum_m \alpha_m(\omega)\mathbf{p}_m, \tag{1}$$

where $\alpha_m(\omega)$ corresponds to the expansion coefficients of mode $\mathbf{p}_m$. These coefficients depend on the overlap of the eigenmode with the external excitation field $\langle \mathbf{p}_m | \mathbf{E}_{\text{exc}} \rangle$ and the eigenvalue $\lambda_m(\omega)$ as

$$\alpha_m(\omega) = \lambda_m(\omega)\langle \mathbf{p}_m | \mathbf{E}_{\text{exc}} \rangle. \tag{2}$$

The two most dominant eigenvector profiles in the expansion are depicted in Fig. 2c, d. From the plot in Fig. 2c, we see a collective dipolar eigenmode $\mathbf{p}_{\text{bright}}$ which is superradiant, whereas in the profile shown in Fig. 2d a subradiant eigenmode $\mathbf{p}_{\text{dark}}$ can be identified. These are exactly the bright and dark modes that are responsible for the Fano interference in our metasystem, and lay the foundation for the relevant physics in the sensing experiment.

We are now in a position to discuss the possible consequences of a perturbation to the modes of a Fano system. First of all, the eigenmode profiles themselves will be perturbed. This is highlighted for both the bright and dark modes in Figs. 2c and d, where the black and red arrows correspond to the unperturbed and perturbed eigenmodes respectively. We note that particularly the dark mode profile shows considerable sensitivity to perturbations. The perturbed eigenmode profiles will in turn lead to an altered overlap of the modes with the external driving field, along with perturbed eigenvalues. These two effects combined consequently lead to perturbed excitation coefficients $\alpha_m(\omega)$, as shown in Fig. 2e, f for both the bright and dark mode, respectively. Here, the absolute value of the complex excitation coefficients is plotted as function of wavelength. Note here that $\alpha_{\text{bright}}(\omega)$ has a relatively weak wavelength dependence, which is expected due to the broadband nature of a mode with large scattering losses. On the other hand $\alpha_{\text{dark}}(\omega)$ shows a strong dispersion and a sharp peak around 780 nm where the Fano interference takes place, and again the largest sensitivity to perturbations. Note that the eigenmode perturbation and corresponding change in overlap with an external driving field is exactly the underlying mechanism that also allows coupling to a symmetry protected quasi BIC resonance, with the perturbation being a structural symmetry breaker ($\Delta L$ here). The prediction of Fig. 2c–f is that this same mechanism brings sensitivity to $\Delta y$.

We have so far discussed the influence of perturbations on eigenmodes only, but not yet on experimental observables with which an experimentalist would actually probe the perturbations. The conventional strategy in sensing is to analyze the spectral response of the Fano system, which would be a single scattering channel reflection or transmission spectrum in the case of nondiffractive metasurfaces, or an angle integrated scattering spectrum in finite sized diffractive structures. In Fig. 2g we show the angle integrated scattering cross section $\sigma_{\text{scat}}$ of the unperturbed and perturbed driven dipole configuration $\mathbf{p}_{\text{driven}}$ for a y-polarized plane wave driving field. A Fano lineshape is evident, resulting from the modal interference of the eigenmodes $\mathbf{p}_{\text{bright}}$ and $\mathbf{p}_{\text{dark}}$. A small shift and decrease in scattering amplitude is also visible for the perturbed cross section, which results from the perturbed excitation coefficient in Fig. 2f. This altered spectral lineshape can be used to for instance translate structural perturbations into measurable intensity differentials when the system is illuminated at resonance, thus providing information about the perturbation at hand. However, as the entire angular spectrum of the scattered signal is integrated, potentially rich angular dependent information is lost in the angle integration.

We now investigate how the perturbation influences directional scattering in Fourier space by resolving the scattered signal over all viewing angles $\hat{\mathbf{k}}$. The acquired Fourier space radiation diagrams

$S_{\text{tot}}(\hat{\mathbf{k}}, \omega)$ can again be decomposed into modal contributions as

$$S_{\text{tot}}(\hat{\mathbf{k}}, \omega) = \alpha_{\text{bright}}(\omega)S_{\text{bright}}(\hat{\mathbf{k}}) + \alpha_{\text{dark}}(\omega)S_{\text{dark}}(\hat{\mathbf{k}}), \tag{3}$$

where $S_{\text{bright}}(\hat{\mathbf{k}})$ and $S_{\text{dark}}(\hat{\mathbf{k}})$ are the modal radiation patterns emitted by $\mathbf{p}_{\text{dark}}$ and $\mathbf{p}_{\text{bright}}$, respectively. Intensity only measurements on a detector can be accounted for by taking the modulus squared of the complex radiation patterns, resulting in

$$|S_{\text{tot}}(\hat{\mathbf{k}}, \omega)|^2 = |\alpha_{\text{bright}}(\omega)S_{\text{bright}}(\hat{\mathbf{k}}) + \alpha_{\text{dark}}(\omega)S_{\text{dark}}(\hat{\mathbf{k}})|^2 = |\alpha_{\text{bright}}|^2|S_{\text{bright}}|^2$$
$$+ |\alpha_{\text{dark}}|^2|S_{\text{dark}}|^2 + 2\text{Re}\,(\alpha_{\text{bright}}\alpha_{\text{dark}}^* S_{\text{bright}}S_{\text{dark}}^*). \tag{4}$$

From this expression it is evident that the total acquired radiation pattern is composed of individual contributions from the bright and dark modes, along with a modal interferometric term. Suppose there is complete transparency in extinction at a certain frequency and that there is no absorption. In that case, the optical theorem states that $\sigma_{\text{scat}} = 0$. Since $\sigma_{\text{scat}}(\omega) \propto \int |S_{\text{tot}}(\hat{\mathbf{k}}, \omega)|^2 d\hat{\mathbf{k}}$, that would mean that the interferometric term in equation (4) fully cancels the first two terms *for every* $\hat{\mathbf{k}}$. By conservation of photon flux it can then be concluded that the modal radiation patterns $S_{\text{bright}}(\hat{\mathbf{k}})$ and $S_{\text{dark}}(\hat{\mathbf{k}})$ should be very similar to each other, and that the expansion coefficients $\alpha_{\text{bright}}(\omega)$ and $\alpha_{\text{dark}}(\omega)$ tune the relative transparency. Perturbations imprinted on the individual eigenmodes would in turn lead to corresponding changes in the modal radiation patterns and, through equation ((4)), on the total radiation pattern $|S_{\text{tot}}(\hat{\mathbf{k}}, \omega)|^2$ as well, which could lead to directional scattering in Fourier space. In Fig. 2h, i, we show Fourier space radiation patterns of the unperturbed and perturbed dipole configurations, while driving the system at resonance at a wavelength of 775 nm. In these figures, we observe concentric rings of bright intensity regions, which correspond to the spatial frequencies or structure factor of the annulus. These rings in k-space are further supplemented by dark intensity regions in the $k_y/k_0$ viewing direction, which are a result of a polarization selective form factor of the collective dipole along the y-direction. The perturbed radiation patterns shows broadly similar features, but marked differences can be spotted with respect to the unperturbed pattern upon closer inspection. To highlight the angular dependent differential signal in Fourier space, we subtract the unperturbed and perturbed Fourier images to arrive at a differential radiation pattern in Fig. 2j. A key advantage of the Fourier space diagram with respect to the angle integrated scattering spectrum in Fig. 2g is that in the former regions with both positive and negative intensity differentials may be resolved, which would otherwise cancel out when the signal would be integrated over all angles. This observation hints towards a potential gain in information about the perturbation when the scattered signal of a Fano resonant structure is dispersed in Fourier space, which we will set out to quantify experimentally in the next sections.

## Spectral resonance characterization

To spectrally characterize the resonant properties of individual dielectric metarings experimentally, we perform dark-field scattering spectroscopy, with a schematic of the setup depicted in the Supplementary Information (SI). In brief, a single metaring (Si meta-atoms on a glass substrate, dimensions and fabrication methods listed in the methods section) is illuminated by a broadband supercontinuum laser with a low effective NA (NA = 0.18), after which the scattered signal is captured in reflection through the same high NA objective (NA = 0.95). Dark-field acquisition is ensured by filtering the low NA back reflection of the illumination out of the total signal by placing a beam stop in the conjugate Fourier plane of the imaging system. Examples of raw spectra are shown in Fig. 3a, where we show a series of

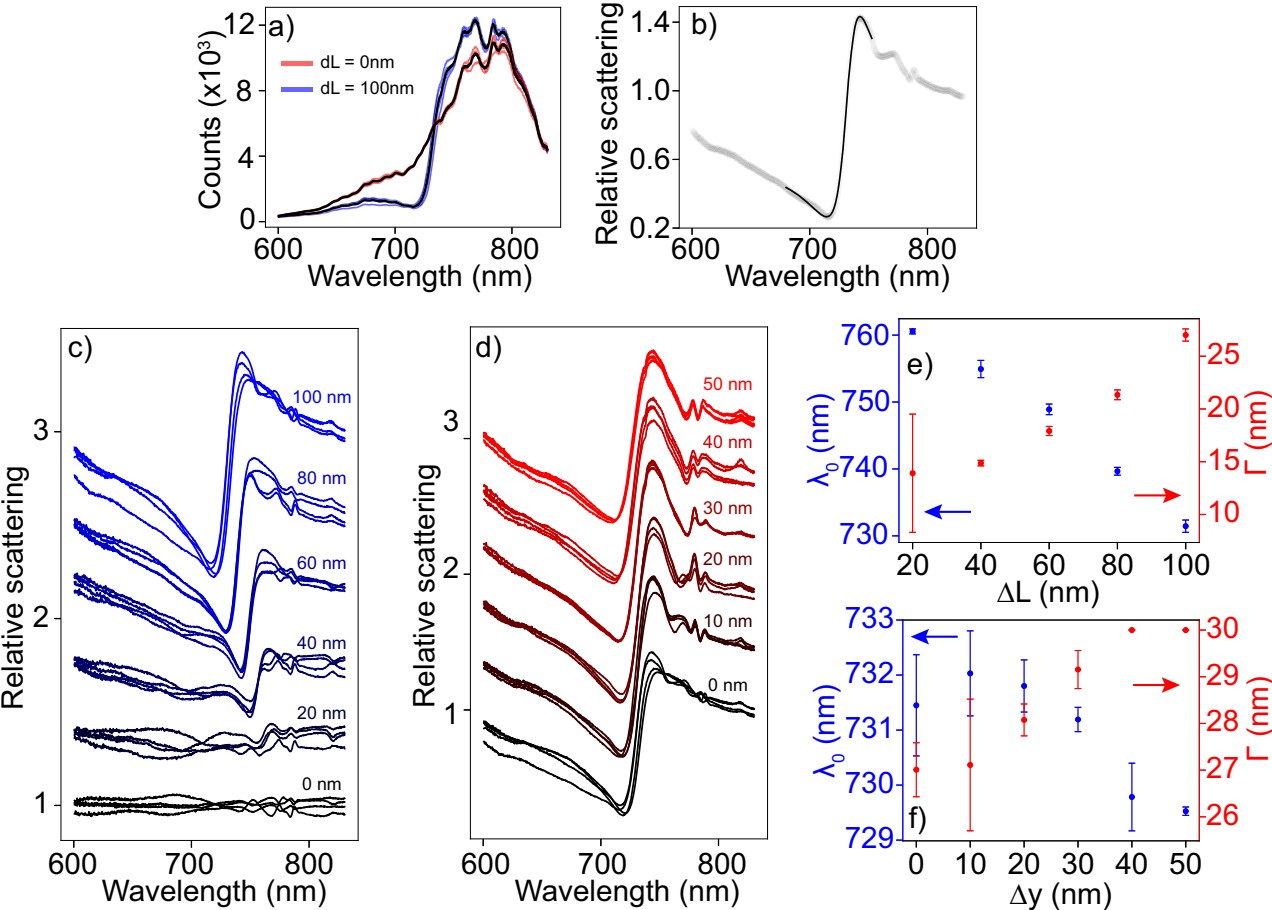

**Fig. 3 | Dark-field spectroscopy on single resonant metarings.** Panel **a**: raw dark-field spectra of individual unperturbed metarings with $\Delta L = 0$ nm (red) and $\Delta L = 100$ nm (blue). The black lines indicate the average spectra. Panel **b**: Example spectrum of a single unperturbed metaring with $\Delta L = 100$ nm normalized by the average spectrum of the $\Delta L = 0$ nm metarings. Shaded datapoints indicate experimental data, the black solid line indicates a fit by a Fano lineshape. Panel **c**: Normalized scattering spectra of unperturbed metarings for increasing asymmetry $\Delta L$. Spectra are offset by 0.4 relative scattering units for clarity. Panel **d**: Normalized darkfield scattering spectra of metarings with fixed asymmetry $\Delta L = 100$ nm for increasing perturbation strength $\Delta y$. Spectra are offset by 0.4 relative scattering units for clarity. Panel **e**: resonance wavelength and linewidth of the unperturbed metarings versus asymmetry $\Delta L$. Panel **f**: resonance wavelength and linewidth of metarings with fixed asymmetry $\Delta L = 100$ nm versus perturbation strength $\Delta y$. Errorbars denote standard deviations of the fit parameters retrieved from the spectra of multiple individual metarings.

darkfield scattering spectra of individual rings with asymmetry parameters of $\Delta L = 0$ nm in red and $\Delta L = 100$ nm in blue, along with their respective averaged spectra in black. Different spectra with the same color indicate spectra scattered by different realizations of metarings with nominally identical fabrication parameters. A good overlap between these individual spectra is seen, indicating that metarings with nominally identical fabrication parameters have reproducible scattering properties. Spectra of the rings with $\Delta L = 100$ nm contain a clear asymmetric feature around 720 nm with respect to the rings with $\Delta L = 0$ nm, hinting towards a Fano lineshape in the scattered signal. To isolate the Fano resonant features within the scattering spectra, we normalize all raw spectra by the average spectrum of the metarings with asymmetry parameter $\Delta L = 0$ nm, as these metarings are not expected to exhibit Fano resonant features due to symmetry protection of the dark mode. In Fig. 3b we show an exemplary normalized spectrum of an individual metaring with asymmetry $\Delta L = 100$ nm. Indeed, an asymmetric Fano like lineshape is seen in the normalized scattering spectrum, and fitting the data in close vicinity around the resonance with a Fano function results in an excellent match. To validate that the Fano feature results from the quasi-BIC in our system, we analyze the normalized scattering

spectra of the metarings as a function of the asymmetry parameter $\Delta L$ in Fig. 3c. We clearly observe a broadening of the resonance linewidth for increasing $\Delta L$, along with a blueshift of the resonance wavelength, both characteristic of a quasi BIC due to symmetry breaking. These observations are substantiated by fit results in Fig. 3e, where we show resonance wavelengths and linewidths extracted by fitting Fano lineshapes to normalized scattering spectra of individual metarings. The datapoints and errorbars in this panel correspond to respectively the mean and standard deviation of the fitparameters retrieved from the spectra of multiple realizations of individual metarings

We next analyze the influence of the perturbation on the scattering behavior, and show the normalized scattering spectra of the metarings with fixed asymmetry $\Delta L = 100$ nm for increasing perturbation strength $\Delta y$ in Fig. 3d. Performing the same fitting analysis on this set of normalized spectra results in extracted resonance wavelengths and linewidths versus $\Delta y$ in Fig. 3f. These results show a broadening of the resonance for increasing $\Delta y$, which can be explained by the fact that the perturbation disturbs the otherwise perfect annular periodicity of the metaring, thereby degrading the resonance quality. This trend is in line with the results of studies on fabrication tolerances on resonant metasurfaces[27]. Apart from the spectral broadening, a

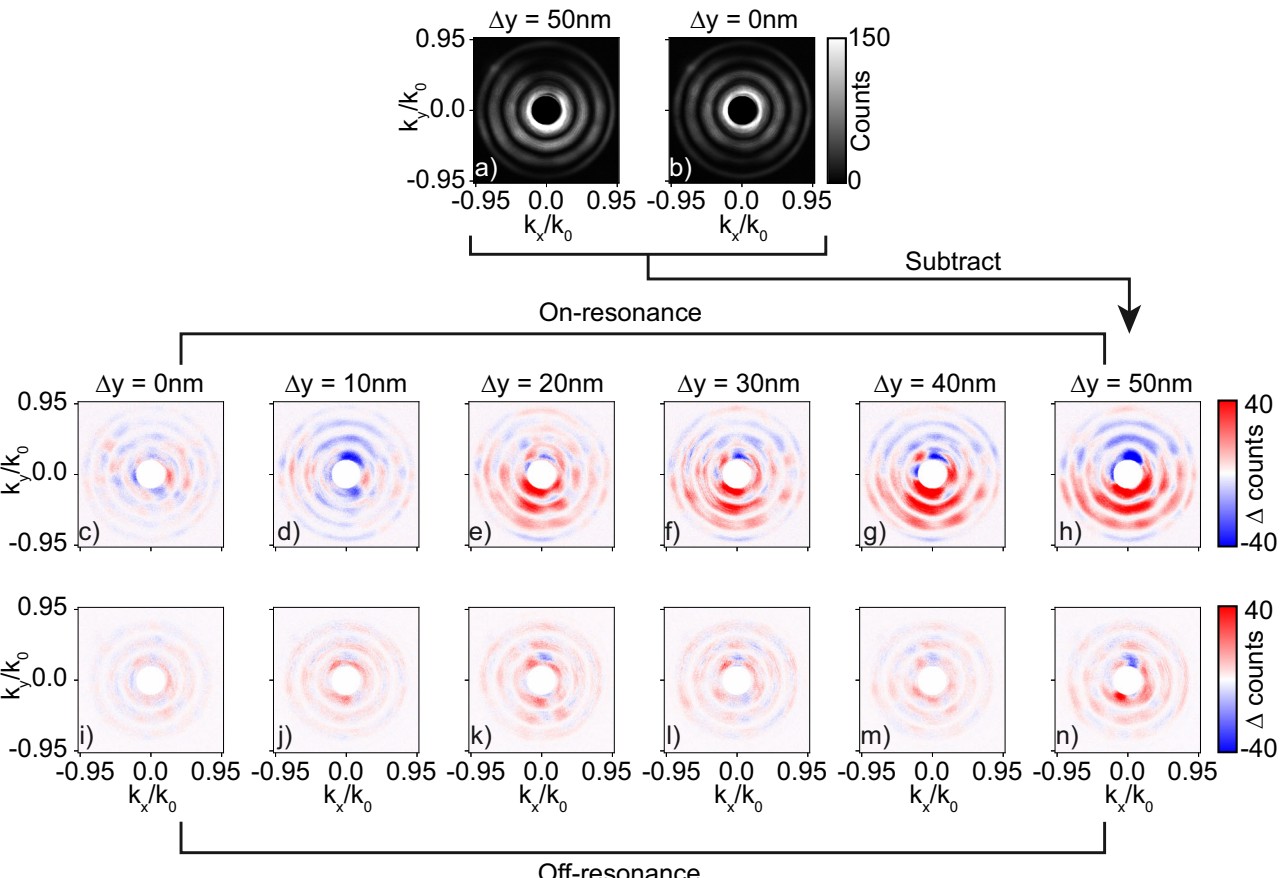

**Fig. 4 | Fourier space scatterometry reveals perturbation induced directional scattering.** Raw images of angle resolved scattering for a metaring with perturbation strengths of panel (**a**): $\Delta y = 50$ nm and panel **b**: $\Delta y = 0$ nm acquired at a wavelength of 730 nm. Angle resolved differential scattering images acquired at a wavelength of 730 nm when the signal of an unperturbed metaring is subtracted from a metaring with perturbation strength of Panel (**c**): $\Delta y = 0$ nm, Panel **d**: $\Delta y = 10$ nm, Panel **e**: $\Delta y = 20$ nm, Panel **f**: $\Delta y = 30$ nm, Panel **g**: $\Delta y = 40$ nm and Panel **h**: $\Delta y = 50$ nm. Panels **i**–**n**: Same as panels (**c**–**h**), but for a wavelength of 780 nm. The results are shown for metarings with asymmetry of $\Delta L = 80$ nm.

minor shift in resonance wavelength is seen as a function of $\Delta y$. These results show that the scattering spectra show deeply subwavelength sensitivity to structural perturbation within the metasurface, albeit that the spectral changes are small.

**Fourier scatterometry**

In this section, we investigate the sensitivity to perturbations of the scattered signal in Fourier space. For this, we utilize Fourier microscopy, (see SI for a sketch of the setup), where we disperse the scattered signal into angular contributions by imaging the back-focal-plane of a microscope objective onto a C-MOS camera. As we did for the spectroscopic measurements described in the previous section, we illuminate individual metarings with a low effective NA (NA = 0.18) and polarization parallel to the y-direction (parallel to the displacement), but now we bandpass filter the excitation wavelength with a bandwidth of 10 nm. We vary the center wavelengths of these filters to spectrally overlap with the Fano resonances as measured in Fig. 2, and capture the corresponding Fourier images scattered by the individual metarings. As such, we acquire directional scattering patterns of individual rings both on and off-resonance, and investigate the sensitivity to perturbations in both spectral regimes.

In Fig. 4a, b, we show exemplary raw Fourier images of perturbed ($\Delta y = 50$ nm) and unperturbed ($\Delta y = 0$ nm) metarings, respectively, with fixed asymmetry parameter ($\Delta L = 80$ nm), acquired at an on-resonance excitation wavelength of 730 nm. In all acquired images, we clip out in software a circle in the center of Fourier space, as it contains

the direct reflection of the illumination by the substrate. The concentric rings originating from the structure factor of the annulus are evident, multiplied by the form factor of the collective dipole emission as we also saw in the calculations of Fig. 2h. We furthermore observe a clear asymmetry in scattered signal in the $k_y/k_0$ viewing direction for the perturbed ring as compared to the scattered signal of the unperturbed ring, indicating that structural perturbations are transduced into observable Fourier space asymmetries. Note that the symmetry axis in both the real and Fourier space is preserved, since displacements in the y-direction in real space lead to specific asymmetries along the $k_y/k_0$ viewing direction in Fourier space.

To better visualize this asymmetry in Fourier space, we subtract the angle resolved scattering patterns of an individual unperturbed ring from a perturbed ring to arrive at a differential Fourier space scattering image. To study the influence of the perturbation strength and the excitation wavelength on the asymmetry in Fourier space, we acquire such differential Fourier images for multiple perturbation strengths and for both on and off-resonance excitation. The results are shown in Fig. 4c–n. From left to right, the perturbation strength is gradually increased from $\Delta y = 0$ nm to $\Delta y = 50$ nm in steps of 10 nm, while in the top and bottom rows the excitation wavelength is set to respectively on (730 nm) and off-resonance (780 nm) excitation. We note that the differential images shown for the two wavelength are scattered by exactly the same pair of rings.

Differential images of unperturbed rings with nominally identical fabrication parameters scatter a relatively low differential contrast-to-

noise ratio in Fourier space as can be seen in Fig. 4c, i. This observation underscores the highly reproducible scattering characteristics of metarings with nominally identical fabrication parameters, as was also evident from the scattering spectra from Fig. 2c. When the perturbation strength is gradually increased from $\Delta y = 0$ nm to $\Delta y = 50$ nm, the differential contrast in Fourier space at resonance in Fig. 4c–h scales accordingly, while the differential contrast off-resonance in Fig. 4i–n remains relatively unaltered. By comparing the calculated differential Fourier space image of Fig. 2j with the images of Fig. 4, we see that a simple dipole model can reproduce the salient features of the experiment. From these measurements, we conclude that deeply subwavelength structural misalignments are resonantly transduced by the Fano resonance into directional scattering asymmetries in Fourier space. It is important to note that the directional scattering effects are sensitive to the polarization state of the incident excitation, since the specific perturbation $\Delta y$ breaks the degeneracy of the bright and dark modes along the x and y direction. An interesting question for future works would be to probe which (polarization) excitation scheme generates maximum information about a given perturbation, which could be tackled with the recently introduced framework of maximum information states[39].

## Information advantage

In the previous two sections, we have presented two different experimental strategies for the detection of a perturbation in a Fano resonant metasystem. The intriguing question which of these two measurements most efficiently detects the perturbation then naturally arises. In this section, we will quantitatively answer this question by determining the information content that each measurement contains about the relevant perturbation. For this, we will use a concept from information theory termed the Fisher information[40], which defines a fundamental bound to the precision with which perturbations $\Theta$ can be estimated from measured data $X$, which are unavoidably corrupted by noise. Assuming that one has access to a theoretical description of the noise in terms of a probability density function $p(X;\Theta)$, which describes the probability of measuring a dataset $X$ given a certain perturbation strength $\Theta$, which in our specific sensing experiment corresponds to structural misalignments $\Delta y$, the Fisher information is defined as[40]

$$F = E\left[\left(\frac{\partial \ln p(X;\hat{\Theta})}{\partial \hat{\Theta}}\right)^2\right], \quad (5)$$

where $E[.]$ denotes the expectation operator. Assuming further that one can construct a so-called unbiased estimator, which is an operation that translates a dataset $X$ into an estimation of the perturbation $\hat{\Theta}$ that on the average should equate the true perturbation $\Theta$, the fundamental precision bound of the estimation is defined by the Cramer-Rao bound

$$\sigma_{\hat{\Theta}}^2 \geq \frac{1}{F}. \quad (6)$$

Here, $\sigma_{\hat{\Theta}}$ is the standard deviation of the estimated parameter $\hat{\Theta}$. From this expression, we see that the higher the Fisher information content, the lower the minimum standard deviation and thus the higher the precision with which the perturbation can be estimated from measured data. Maximization of the Fisher information has already been shown to positively impact the measurement precision in experimental contexts such as 3D localization and imaging[41], sensing perturbations in disordered environments[39,42,43] and tracking single molecules[44].

To arrive at a closed form expression for the Fisher information in the context of our sensing experiment, we will now discuss the several noise factors that can corrupt our measured data and thus reduce the

available Fisher information. The first noise source originates from the unavoidable fabrication errors that invoke additional shape perturbations besides the controlled perturbation that we want to detect. Although our structures show decent reproducibility in scattering properties, two individual metasystems with nominally identical fabrication parameters will never scatter in a perfectly identical way due to fabrication imperfections, as could already be observed from Figs. 4c and i, where small residues in scattering contrasts exist above the inherent noise floor of the measurement. The measurement noise floor in turn consists of photon shot noise and detector read noise, which we describe in the (SI). For simplicity, we have assumed that the only noise sources originate from photon shot noise and detector related noise sources in the derivation of the Fisher information, and describe how we correct for the fabrication noise later on in this section. For analytical tractability, we describe the noise in a single pixel on the detector as Gaussian

$$p(X;\hat{\Theta}) = \frac{1}{\sqrt{2\pi\sigma_{tot}^2(\hat{\Theta})}} e^{-\frac{(X - I_0(\hat{\Theta}))^2}{2\sigma_{tot}^2(\hat{\Theta})}}, \quad (7)$$

which represents the probability of measuring $X$ photons given a mean number of measured photons $I_0(\hat{\Theta})$ and a standard deviation given by the sum of shot-noise and detector readout noise $\sigma_{read}$ contributions

$$\sigma_{tot}^2(\hat{\Theta}) = I_0(\hat{\Theta}) + \sigma_{read}^2. \quad (8)$$

The choice of a Gaussian distribution instead of Poissonian is a convenient way to account for the mixed contributions from shot noise and detector readout noise, and for the count rates at hand the Gaussian distribution approaches the Poissonian one. Using this Gaussian noise model we can therefore make reasonable estimations of the available Fisher information in the data by using a closed form analytical expression, which is obtained by substituting equations (7) and (8) into equation (5):

$$F = \frac{2(I_0 + \sigma_{read}^2) + 1}{2(I_0 + \sigma_{read}^2)^2}\left(\frac{\partial I_0}{\partial \hat{\Theta}}\right)^2. \quad (9)$$

Assuming that we can approximate the differential in Eq. (9) with finite differences as $\frac{\partial I_0}{\partial \hat{\Theta}} = \frac{I(\Delta y) - I(0)}{\Delta y}$ and $I_0$ as the scattered intensity of an unperturbed ring $I(0)$, we can estimate the Fisher information content of a sensing experiment from an unperturbed and perturbed measurement, for which we already acquired all the relevant data in the previous sections. Besides analytics, we note that model-free approaches exist to determine the available Fisher information from data where the noise model is inherently unknown[45].

We now quantitatively compare the Fisher information content between an angle integrated scattering measurement and Fourier space measurement. To keep all experimental parameters such as integration time, incident laser power and detector efficiency a constant in the comparison, we estimate the Fisher information in the two different measurement strategies from our Fourier space data alone, constructing spectral data by software integration over angle. Figure 5 summarizes the workflow to arrive at quantitative values for the Fisher information content. We start with raw Fourier images scattered by an unperturbed ($\Delta y = 0$ nm) and perturbed ring ($\Delta y = 50$ nm) with fixed asymmetry ($\Delta L = 80$ nm) and excited by a wavelength of 730 nm in Fig. 5a, b, respectively. Following the red connector lines, from these two images we calculate an angle resolved Fisher information image by applying Eq. (9) onto the individual non-zero count pixels of the images, with the result depicted in Fig. 5c. From this image we can see that Fisher information is directionally scattered in Fourier space, with the majority of the information scattering into the negative $k_y/k_0$ viewing direction. This can be explained by again inspecting Fig. 4h, which is

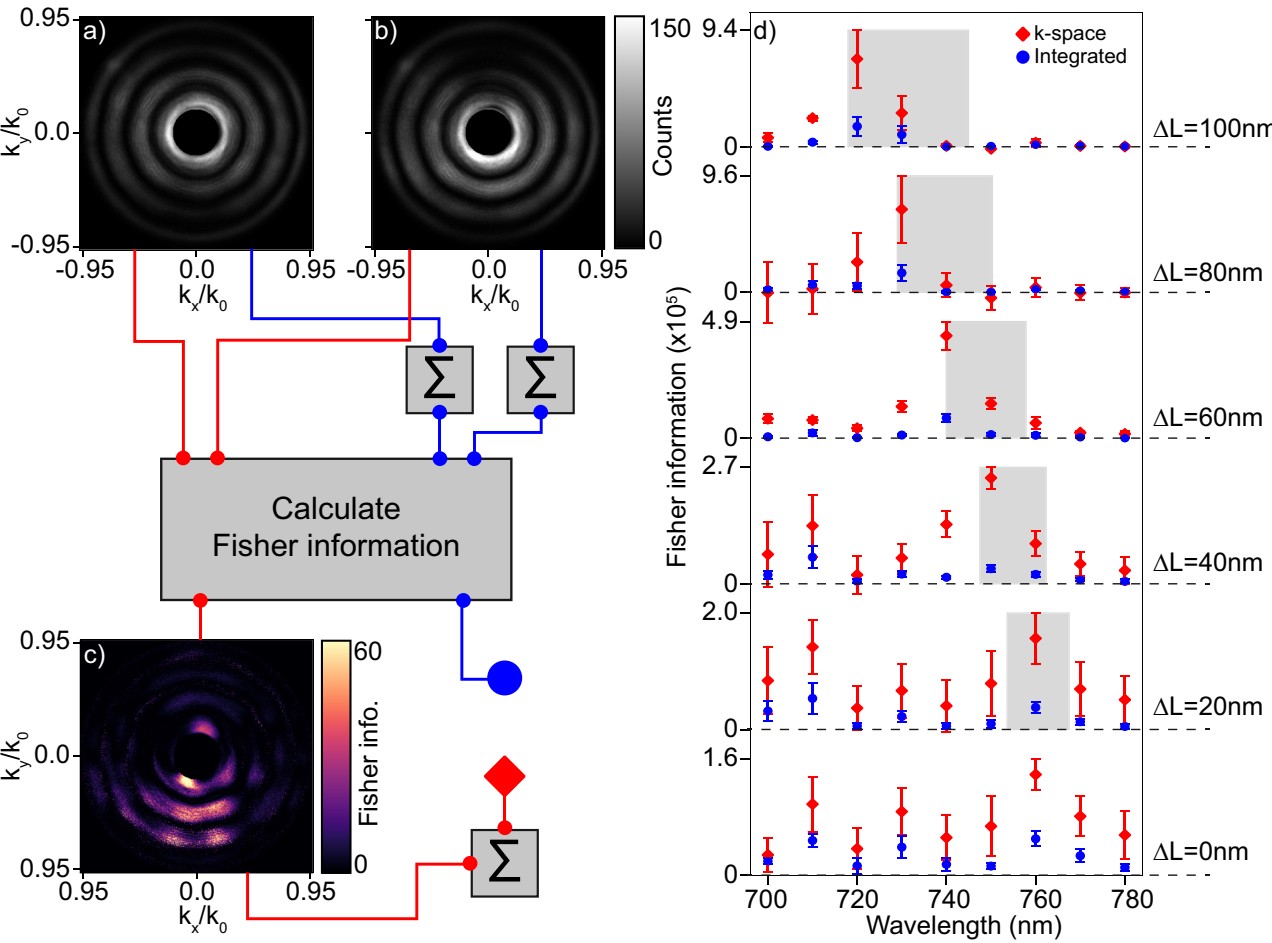

**Fig. 5 | Information comparison between angle integrated and angle resolved scattering measurements.** Panels **a** and **b** show raw data of unperturbed ($\Delta y = 0$ nm) and perturbed ($\Delta y = 50$ nm) Fourier space images for a metaring with fixed asymmetry of $\Delta L = 80$ nm, excited at a wavelength of 730 nm. These two images are processed through the red and blue networks in the figure to arrive at an information value for the angle resolved and angle integrated measurements respectively. Panel **c** shows the angle resolved Fisher information resulting from the pair of measurements of panels (**a**) and (**b**). Panel **d** shows the total Fisher information of angle integrated and angle resolved measurement as blue circles and red diamonds respectively, plotted versus excitation wavelength and for all realizations of the asymmetry $\Delta L$. Datapoints and errorbars denote mean and standard deviation of a set of 4 similar individual metaring pairs. The center and width of the gray bars denote resonance wavelength and linewidths extracted from dark-field spectra of unperturbed rings.

the differential intensity image that corresponds to the Fisher information radiation pattern as shown in 5c, from which we can see that indeed the majority of the differential counts are scattered into the negative $k_y/k_0$ direction. A recent work[46] has shown that the shape of these information patterns can be understood from a near-field description using so-called information sources, which radiate information instead of energy, and poses the interesting design question how antenna theory could be exploited to engineer these information patterns at will. To connect a single information value to this radiation pattern, we make use of the additivity of the Fisher information and sum all the values in the Fisher information radiation pattern.

The retrieved value for the Fisher information will be generated by the intentional perturbation $\Delta y$, but also by unintentional nanoscale perturbations due to imperfections in nanofabrication $\xi$. To see how $\xi$ can generate information in Fourier space, we again turn to Fig. 4c, i, where it can be seen that 2 meta rings with the same fabrication parameters still generate a differential count contrast due to the fabrication noise $\xi$, which will be added to the total Fisher information. To isolate the information that is generated purely from the intentional perturbation $\Delta y$ that we want to probe, we apply the same workflow of Fig. 5a–c on meta ring pairs with similar fabrication parameters (ring 1: $\Delta y = 0$ nm, ring 2: $\Delta y = 0$ nm). The corresponding Fisher information $F_\xi$

defines the information generated by fabrication noise $\xi$. The Fisher information generated purely by $\Delta y$ is then isolated by subtracting the information generated purely from fabrication noise from the information generated by the combination of $\xi$ and $\Delta y$ as $F_{\Delta y} = F_{\Delta y + \xi} - F_\xi$, which finally represents the red diamond datapoint in Fig. 5.

To estimate the information content of a simple spectral measurement (blue network) we first sum the individual Fourier space images in Figs. 5a, b. This operation simulates the measured count value a single pixel on a spectrometer would detect for the relevant excitation wavelength. We finally apply equation (9) onto this pair of values to arrive at the final information estimate represented by the blue circle. In Fig. 5d, we showcase the full comparison in information content between an angle integrated and angle resolved measurement. For this, we determined Fisher information values for a fixed perturbation strength of $\Delta y = 50$ nm, while varying the excitation wavelength and the asymmetry $\Delta L$. We have used Fourier scatterometry data of four independent meta ring pairs to determine average and standard deviations of the information content of both measurement modalities, indicated by datapoints and errorbars respectively. In the same figure, the center and width of the gray vertical bars indicate the resonance wavelengths and linewidths of unperturbed metarings respectively. It can be seen that the information content of both the

angle resolved and angle integrated measurements peak at the Fano resonance, where the peak follows the same blue shifted trend versus increasing asymmetry $\Delta L$ as we have observed in the spectroscopic measurements of Fig. 2, corroborating the fact that sensing at resonance more efficiently detects a perturbation than sensing off-resonance from an information perspective. We further see that the information content of a Fourier space measurement is always higher than that of an angle integrated measurement, both on and off-resonance, where the information gain can be as high as sevenfold. From the scalebar we can finally observe that the maximum information content increases with increasing asymmetry $\Delta L$, ultimately saturating beyond $\Delta L = 80$ nm. To clarify this behavior, we recall that the Fisher information performance metric is not only influenced by conventional factors such as resonance wavelength shifts and Q-factor, but also by the resonant amplitude that defines the molulation depth of the Fano lineshape. Indeed, a recent study[47] has demonstrated that optimal sensing performance does not necessarily correspond to the scenario of highest Q-factor, but rather to balancing the Q-factor and resonance amplitude. Since the resonance amplitude increases versus increasing $\Delta L$, as evident from Figure 3c, a similar argument can be applied to the meta ring scattering sensor. Furthermore, the novelty of Fourier space analysis of resonant scattering requires a new performance metric that quantifies the degree of directional scattering induced by a perturbation, which would be an interesting topic for future investigations. All these factors combined influence the ultimate Fisher information plotted in Fig. 5d and are responsible for the saturation in Fisher information beyond $\Delta L = 80$ nm.

We finally note that the information advantage of a Fourier space measurement with respect to an angle integrated measurement is generally valid for other Fano-resonant structures and perturbation types beyond those investigated in this paper. The crucial necessity is that the photonic structure should be able to scatter light into higher-order diffraction channels. The generality of the approach can be understood by viewing the Fano-resonance not only as a near-field coupling effect between bright and dark modes: the optical theorem fundamentally implies that any near-field effect that gives rise to a Fano line also expresses as a far-field modal interferogram. This interferometric advantage can generate specific sensitivities as a function of viewing angle due to perturbations. To support this claim, we have performed coupled dipole calculations on different Fano-resonant structures and perturbation types in the SI, all of which demonstrate the information advantage of Fourier space readout. Since Fisher information (in shot-noise limited experiments) scales with the number of detected photons, a similar information content can be retrieved from a Fourier space measurement compared to conventional spectroscopy, yet for a lower photon budget. This means that the integration time, or incident power could be reduced in a Fourier space read out scheme, while not losing Fisher information with respect to conventional spectroscopy.

## Discussion

In summary, we have studied deeply subwavelength perturbations in Fano resonant dielectric metasurfaces, and experimentally demonstrated that an information gain in resonance based sensing experiments can be obtained when analyzing the scattered signal of the metasystem in Fourier space. We observed that the information gain is a consequence of resonant directional scattering of the metasurface induced by a perturbation, a feature that is not captured in conventional resonance based sensing schemes, and have theoretically discussed possible origins of this directional scattering by examining how a perturbation affects the system's fundamental eigenmodes. We quantitatively determined the information gain by calculating the Fisher information contents of both an angle resolved and angle integrated detection scheme assuming shot-noise and detector read-out noise as dominant noise sources in the experiment. We estimated

that an information gain as high as a factor of 7 can be achieved using the Fourier space sensing modality, which means that the exposure time or incident laser power can be reduced by this same factor to achieve similar sensing precisions as in conventional spectral based sensing protocols. This is appealing in sensing experiments where photon budget[44] and fast measurement times[35] play an important role.

Looking forward, we speculate that additional information about a perturbation can be retrieved by adding polarizing optics and interferometric tools in the detection setup to retrieve the full polarization state along with the phase of the scattered signal, such that a complete description of the scattered response is acquired. Besides, one could further control the external degrees of freedom on the excitation side by, for instance, polarization[22] and wavefront[39] shaping to achieve an optimally informative illumination. Such control, in combination with designer metastructures, might also open pathways towards arbitrary design of the information flow in Fourier space[46]. Finally, estimators that reach the Cramer-Rao bound can be constructed to perform actual estimations of $\Delta y$ by analysis of the Fourier scatterometry images. For this, artificial neural networks have been shown to efficiently reach the detection limit set by the Fisher information as determined in this work[45]. All in all, our work explored advanced sensing strategies that leverage the full potential of resonant metasurfaces, paving the way for more efficient, precise, and versatile sensing platforms.

## Method
### Coupled dipole calculations
The metarings discussed in the main text are theoretically modeled by coupled point dipoles. For an isolated dipole, we use the dyadic Green's function[38] of a dipole $\mathbf{p}$ positioned at $\mathbf{r}_0$ in a homogeneous medium to describe the scattered fields at a probing position $\mathbf{r}_1$ as:

$$\ddot{\mathbf{G}}(\mathbf{r}_0, \mathbf{r}_1) = \left( \vec{\mathbf{I}} + \frac{1}{k^2} \boldsymbol{\nabla}\boldsymbol{\nabla} \right) \frac{e^{ik|\mathbf{r}_1 - \mathbf{r}_0|}}{4\pi|\mathbf{r}_1 - \mathbf{r}_0|}, \tag{10}$$

such that

$$\mathbf{E}(\mathbf{r}_0, \mathbf{r}_1) = \omega^2 \mu\mu_0 \ddot{\mathbf{G}}(\mathbf{r}_0, \mathbf{r}_1) \cdot \mathbf{p} \tag{11}$$

Here, $\vec{\mathbf{I}}$ is the unit dyad, $\omega$ is the driving frequency, $k = n\frac{w}{c}$ is the wavenumber in the medium, $\mu$ the permeability of the medium and $\mu_0$ the free space permeability. For an ensemble of N coupled dipoles $\{\mathbf{p}_1...\mathbf{p}_N\}$ driven by an incident electric field $\mathbf{E}_{\text{exc}}$, the solution of the n'th dipole configuration can be described by a self consistent set of equations as:

$$\mathbf{p}_n = \alpha_n \left( \mathbf{E}_{\text{exc}}(\mathbf{r}_n) + \sum_{m \neq n} \ddot{\mathbf{G}}(\mathbf{r}_n, \mathbf{r}_m) \cdot \mathbf{p}_m \right), \tag{12}$$

where $\alpha_n$ is the polarizability of the n'th dipole. This system of equations can be recast into a matrix form as

$$\begin{bmatrix} \mathbf{p}_1 \\ \vdots \\ \mathbf{p}_N \end{bmatrix} = M^{-1} \cdot \begin{bmatrix} \mathbf{E}_{\text{exc}}(\mathbf{r}_1) \\ \vdots \\ \mathbf{E}_{\text{exc}}(\mathbf{r}_N) \end{bmatrix}, \tag{13}$$

and

$$M_{i,j} = \delta_{i,j}\alpha_i^{-1} - (1 - \delta_{i,j})\ddot{\mathbf{G}}(\mathbf{r}_0, \mathbf{r}_1). \tag{14}$$

Here, the $(3N \times 3N)$ matrix $M$ is the coupling matrix of the system, and describes the interactions between all the dipoles in the system. The scattered fields of the ensemble of dipoles, and the optical quantities that result from this can thus be calculated by a matrix inversion of the coupling matrix $M$. We analytically model the static polarizabilities of

the dipolar particles in the metarings as prolate spheroids[48], with a long axis $L_0$ of 300 nm and short axis $W$ of 100 nm. These particles are positioned in an annulus of radius 1.5 μm, and every other particle in the annulus is reduced in length by $L_0 - \Delta L$, where $\Delta L = 100$ nm. The dielectric permittivity of the particles $\epsilon$ is set to 12, approximating that of silicon, while the surrounding medium is set to vaccum $\epsilon_m = 1$. Radiation damping[49] is included to the static polarizabilities $\alpha_0$ as $\frac{1}{\alpha} = \frac{1}{\alpha_0} - i\frac{1}{6\pi\epsilon_0}k^3$. Finally, we compute the eigenvectors and corresponding eigenvalues of the coupling matrix $M$ to retrieve the eigenmodes of the coupled system.

## Sample design and nanofabrication

The dielectric metarings in the main text were fabricated using the following procedure. A glass coverslip (Menzel-Gläser) with a thickness of 170 μm was sonicated for 10 minutes in $H_2O$ and cleaned in a solution of base piranha ($NH_4OH/H_2H_2/H_2O = 1/1/5$) at 75 °C for 15 min. After thoroughly rinsing excess piranha with $H_2O$, the sample was blown dry with $N_2$. A 120 nm layer of amorphous silicon (a-Si) was evaporated onto the cleaned coverslips using electron beam evaporation (Polyteknik Flextura M508 E) at a deposition rate of 0.1 nm/s. After an oxygen plasma descum process of 2 min, a 60 nm layer of hydrogen silsequioxane (HSQ) was spincoated (CEE Apogee 200 Spin Coater) for 60 s with a speed of 8000 rpm and acceleration of 1000 rpm/s and baked for 2 min at 180 °C. A conductive aluminum layer of 10 nm was thermally evaporated (Polyteknik Flextura M508 E) at a rate of 0.05 nm/s. For alignment purposes of the electron beam (Raith), 50 nm gold nanoparticles were dropcasted in the corners of the substrate, after which the sample was exposed at 50kV acceleration voltage and an area dose of 1500 μC/cm2. After exposure, the substrate was chemically developed in a solution of tetramethylammoniumhydroxide (TMAH) for 60 s at 60 °C and rinsed with $H_2O$ for 15 s. Finally, the substrate was etched (Oxford Plasmalab 80+) using a mixture of 10 sccm sulfurhexafluoride ($SF_6$), 15 sccm trifluoromethane ($CHF_3$) and 3 sccm $O_2$. The length, width and height of the ($\Delta L = 0$) rods in the ring corresponds to 335 nm, 115 nm and 120 nm, respectively.

## Data availability

All the source data supporting the findings of this study are available within a replication package stored on the public repository ZENODO at https://doi.org/10.5281/zenodo.17583238[50].

## Code availability

All the code used to produce the results and figures presented in this work are available within a replication package stored on the public repository ZENODO at https://doi.org/10.5281/zenodo.17583238[50].

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

## Acknowledgements

This work is part of the Dutch Research Council (NWO) and was performed at the research institutes ARCNL and AMOLF. The Advanced Research Center for Nanolithography ARCNL is a public-private partnership between the University of Amsterdam, Vrije Universiteit Amsterdam, Rijksuniversiteit Groningen (RUG), the Netherlands Organization for Scientifc Research (NWO), and the semiconductor-equipment manufacturer ASML.

## Author contributions

N.F. and A.F.K. conceived the project idea. N.F. performed theoretical modeling, sample fabrication, measurements, interpretation, and wrote the first draft of the manuscript. A.J.d.B. contributed to discussions and interpretation. L.V.A. and A.F.K. contributed to discussions, interpretation, writing and overall supervision.

## Competing interests

The authors declare no competing interests.
