## [Transparent Peer Review file · Nature Communications]

Information advantage in sensing revealed by Fano resonant Fourier scatterometry

Corresponding Author: Professor Femius Koenderink

Version 0:

Reviewer comments:

Reviewer #1

(Remarks to the Author)

The paper is based on several known concepts embedded into a new envelope shaped around "information advance in sensing". In reality, the authors employ the so-called radial quasi-BIC modes (introduced and demonstrated earlier in Ref. 33 but termed here "Fano resonances") for sensing (also demonstrated earlier, including the similar systems) and discuss how a perturbation affects fundamental properties of resonant systems. More specifically, they show that perturbations can lead to marked directional scattering in the Fourier space.

The real novelty of this paper is a comparison of two sensing methods in terms of their inherent Fisher information content, but this message and related results are not strong enough to justify publication in a highly ranked Nature Communications.

(Remarks on code availability)

Reviewer #2

(Remarks to the Author)

This manuscript presents significant research elucidating, both theoretically and experimentally, the superior information acquisition capability of Fourier space analysis over conventional spectral analysis in sensing with Fano resonant metasurfaces. The introduction of Fisher information to quantitatively demonstrate this "information advantage" shows potential for proposing a new analytical paradigm in the sensing field. The study design is systematic, and the experimental results robustly support the claims. However, further clarification and discussion on a few points would enhance the paper's completeness and impact.

This work is judged to have sufficient potential for publication in Nature Communications after addressing the following concerns:

1. The method described for correcting fabrication noise needs a clearer explanation of the statistical basis and specific calculation steps for subtracting the "corresponding result." This elaboration on fabrication noise correction in Fisher Information Calculation is critical for the reliability of the results and requires detailed description.
2. To bolster the current "speculate" statement, either add more convincing discussion by analyzing parameters like Q-factor, scattering intensity, and resonance shift sensitivity with varying ΔL , or clearly state this as a subject for future investigation. Strengthening the justification for maximum information gain at $\Delta L=80$ nm would improve the manuscript.
3. The direct interpretation and discussion of how theoretical eigenmode changes in Figure 2 manifest as experimental Fourier space asymmetries in Figure 4 should be strengthened. Specifically, an explicit explanation linking the perturbation of the pdark mode to the observed directional scattering is needed to enhance the connection between theory and experiment.
4. The apparent author's note "Radiation damping?" in the Methods Section should be removed or revised into a complete statement.
5. A brief consideration of whether the proposed methodology is a general principle applicable to other types of Fano resonant systems or perturbations beyond the specific system studied would be welcome when discussing the general applicability of Fourier space analysis.

6. Consider adding brief explanations for terms like "interlaced metrology" that might be unfamiliar to readers outside that specific field to add a terminology explanation.
7. The logical connection between the cancellation by the interference term and the similarity of modal radiation patterns should be clarified to reinforce the logic in Equation (4) discussion.
8. Enhance the readability of the optical component list by adjusting text size or explicit mention in the caption to improve the readability of Figure S1 in Supplementary Information.
9. A sentence or two commenting on the validity of the Gaussian noise assumption for Fisher information calculation would be appropriate when briefly discussing the Gaussian noise assumption.

(Remarks on code availability)

Reviewer #3

(Remarks to the Author)

This manuscript presents a study demonstrating that angle-resolved measurements of light scattered from Fano-resonant dielectric metasurfaces provide significantly more information than conventional spectroscopic (angle-integrated) detection strategies. The authors quantify this improvement using Fisher information theory and validate the concept theoretically and experimentally.

I find this work to be well executed in both its theoretical and experimental parts, and exploiting the angular (Fourier-space) information content of the far field appears to be a novel concept in resonant nanophotonic sensing, going beyond typical sensing schemes involving only the spectral resonance shift. The finding of a sevenfold increase in information demonstrates the advantage of this approach (under idealized conditions).

In the following section, I list a few points for the authors to consider:

- + What are the prospects of generalising this approach to other common Fano-resonant structures?
- + Both the perturbation and the excitation polarisation are aligned along the y-axis. How sensitive are the results with respect to this alignment? Would using different polarizations affect the directional sensitivity?
- + Can the authors provide a prospect on the practical applications of their work? Could it be useful in semiconductor processing or biosensing with fewer photons, for example?
- + Although this probably goes beyond the scope of this work, it could also be interesting to see if an artificial neural network can reach the detection limit resulting from the Fisher information analysis in this work (along the lines of Nature Photonics 19, 593–600 (2025)).

In summary, I think this is an interesting piece of work that should be published in Nature Communications once the above suggestions have been taken into account.

(Remarks on code availability)

Version 1:

Reviewer comments:

Reviewer #1

(Remarks to the Author)

The authors made a very serious revision and addressed clearly my major concerns. I did like that they extended the study by adding other types of Fano-resonant structures and perturbations that are frequently employed by the sensing community. This new section "Different Fano structures and perturbation types" added to the supplementary part is very useful, and it covers different scenarios of Fano resonant structures and perturbation types.

I am happy to suggest publication in the current form

(Remarks on code availability)

Reviewer #2

(Remarks to the Author)

As a reviewer specializing in devices utilizing Fano resonances, I would like to clarify that my expertise lies primarily in the development and characterization of such photonic structures, rather than in the field of sensing methodologies or information theory. While I appreciate the authors' efforts to expand the sensing paradigm through the analysis of Fisher

information in the Fourier space, I am not in a position to critically evaluate the novelty or impact of this approach from a sensing expert's perspective.

(Remarks on code availability)

Reviewer #3

(Remarks to the Author)

This manuscript NCOMMS-25-26949A has been seen by altogether three reviewers, who provided comments and suggestions.

From my point of view, the authors have carried out a thorough revision of their manuscript and have also provided a very detailed rebuttal letter in which all concerns by the referees have been addressed. Apart from a number of technical clarifications, the authors have also addressed the concern of Referee 1 regarding novelty: I strongly agree here with the authors, that the information stored in a Fourier space analysis clearly goes beyond an analysis of the spectral lineshape only.

In light of the above, I strongly recommend this manuscript for publication in Nature Communications.

Minor points to clarify at the proof stage:

+ In line 434 there is a typo "...does not necessarily corresponds..."

+ Typically the definition of the S-matrix does not only involve the scattered part, but also the incident part (in contrast to the transition matrix T). In this sense, the scattering cross section depends only on the scattered field, not on the incident one (see expressions below Eq. (4)). I assume this is just an issue of convention though, where the authors denote S_{tot} to be the matrix that describes "total radiated (scattered) field from all modes" rather than "total field = incident + scattered field", correct? But maybe I also just misunderstood something here.

(Remarks on code availability)

Reviewer 1

Reviewer comment

The paper is based on several known concepts embedded into a new envelope shaped around “information advance in sensing”. In reality, the authors employ the so-called radial quasi-BIC modes (introduced and demonstrated earlier in Ref. 33 but termed here “Fano resonances”) for sensing (also demonstrated earlier, including the similar systems) and discuss how a perturbation affects fundamental properties of resonant systems. More specifically, they show that perturbations can lead to marked directional scattering in the Fourier space.

The real novelty of this paper is a comparison of two sensing methods in terms of their inherent Fisher information content, but this message and related results are not strong enough to justify publication in a highly ranked Nature Communications.

Authors' response

We thank reviewer 1 for their time to read the manuscript and provide feedback. We would like to stress that we do not claim that Fano resonances per se, or radial quasi BIC modes, are novel for sensing. Instead the claim is that Fourier analysis is fundamentally quantitatively superior to spectral sensing. Fourier space analysis of the scattering of a Fano resonant structure has, to the best of our knowledge, hitherto never been considered in the sensing community, where perturbations have so far been determined by analyzing distortions in spectral lineshapes only. The superior information content of Fourier space analysis compared to conventional analysis of spectral lineshapes constitutes a *general* result, and is thereby valid for other Fano resonant structures and perturbation types beyond the specific meta ring geometry discussed in the main text. This is supported by revisions and additional calculations.

Modifications

To improve the message regarding the generality of our approach, we have added the following paragraph on page 24, starting from line 443:

“We finally note that the information advantage of a Fourier space measurement with respect to an angle integrated measurement is generally valid for other Fano-resonant structures and perturbation types beyond those investigated in this paper. The crucial necessity is that the photonic structure should be able to scatter light into higher-order diffraction channels. The generality of the approach can be understood by viewing the Fano-resonance not only as a near-field coupling effect between bright and dark modes: the optical theorem fundamentally implies that any near-field effect that gives rise to a Fano line also expresses as a far-field modal interferogram. This interferometric advantage can generate specific sensitivities as a function of viewing angle due to perturbations. To support this claim, we have performed coupled dipole calculations on different Fano-resonant structures and perturbation types in the SI, all of which demonstrate the information advantage of Fourier space readout.”

We have also extended our study towards other types of Fano-resonant structures and perturbation types that are frequently encountered in the sensing community. To this end, we have added a new section “Different Fano structures and perturbation types” to the supplementary information in which we theoretically investigate the information advantage of Fourier space analysis over angle integrated spectral analysis for different scenarios of Fano resonant structures and perturbation types.

Reviewer 2

Reviewer comment

This manuscript presents significant research elucidating, both theoretically and experimentally, the superior information acquisition capability of Fourier space analysis over conventional spectral analysis in sensing with Fano resonant metasurfaces. The introduction of Fisher information to quantitatively demonstrate this "information advantage" shows potential for proposing a new analytical paradigm in the sensing field. The study design is systematic, and the experimental results robustly support the claims. However, further clarification and discussion on a few points would enhance the paper's completeness and impact.

Authors' response

We appreciate the positive remarks of reviewer 2 on our work and would like to thank them for their detailed review and comments. In the following, we will address all comments of the reviewer in detail, together with the associated modifications made to the manuscript.

Reviewer comment

This work is judged to have sufficient potential for publication in Nature Communications after addressing the following concerns:

1. The method described for correcting fabrication noise needs a clearer explanation of the statistical basis and specific calculation steps for subtracting the "corresponding result." This elaboration on fabrication noise correction in Fisher Information Calculation is critical for the reliability of the results and requires detailed description.

Authors' response

The Fisher information that is generated in an angle resolved measurement originates from both the intentional perturbation Δy , but also from unintentional nanoscale perturbations ξ due to imperfections in nanofabrication. To isolate the information that is generated purely from the intentional perturbation Δy , we apply the workflow of Figure 5a-c on meta ring pairs with similar fabrication parameters (ring 1: $\Delta y = 0$ nm, ring 2: $\Delta y = 0$ nm). The corresponding Fisher information F_{ξ} defines the information generated by fabrication noise ξ . Next, we apply the same workflow on meta ring pairs with different values of Δy (ring 1: $\Delta y = 50$ nm, ring 2: $\Delta y = 0$ nm) as described in the main text, where the retrieved Fisher information $F_{\xi+\Delta y}$ will be generated by both ξ and Δy . The Fisher information generated purely by Δy is then determined by $F_{\Delta y} = F_{\xi+\Delta y} - F_{\xi}$. This procedure is repeated for 4 different meta ring pairs, and datapoints and errorbars in Figure 5d denote mean and standard deviations over these 4 ring pairs. We agree that a more elaborate explanation is justified, and have incorporated the changes outlined below.

Modifications

To better clarify the procedure for fabrication noise correction, we added the following paragraph on page 22, line 396:

“The retrieved value for the Fisher information will be generated by the intentional perturbation Δy , but also by unintentional nanoscale perturbations due to imperfections in nanofabrication ξ . To see how ξ can generate information in Fourier space, we again turn to Figure 4c and 4i, where it can be seen that 2 meta rings with the same fabrication parameters still generate a differential count contrast due to the fabrication noise ξ , which will be added to the total Fisher information. To isolate the information that is generated purely from the intentional perturbation Δy that we want to probe, we apply the same workflow of Figures 5a-5c on meta ring pairs with similar fabrication parameters (ring 1: $\Delta y = 0$ nm, ring 2: $\Delta y = 0$ nm). The corresponding Fisher information F_ξ defines the information generated by fabrication noise ξ . The Fisher information generated purely by Δy is then isolated by subtracting the information generated purely from fabrication noise from the information generated by the combination of ξ and Δy as $F_{\Delta y} = F_{\xi+\Delta y} - F_\xi$, which finally represents the red diamond datapoint in Figure 5.”

To explain the statistical origins of the errorbars in figure 5d, we have added the following sentence on page 23, line 417:

“We have used Fourier scatterometry data of four independent meta ring pairs to determine average and standard deviations of the information content of both measurement modalities, indicated by datapoints and errorbars respectively.”

Reviewer comment

2. To bolster the current "speculate" statement, either add more convincing discussion by analyzing parameters like Q-factor, scattering intensity, and resonance shift sensitivity with varying ΔL , or clearly state this as a subject for future investigation. Strengthening the justification for maximum information gain at $\Delta L=80$ nm would improve the manuscript.

Authors' response

The Fisher information quantifies the performance of our resonant sensor by relating an actual change in detected intensity to the noise floor of the measurement scheme. The change in signal in our experiment is influenced by factors encapsulated in the conventional sensing FOM, which is the wavelength shift per unit perturbation multiplied by the resonance Q-factor, typically stating that the best sensing performance is reached with highest possible resonant Q-factor. This performance merit, however, does not take the noise of the measurement into account. Equally important in our case is the aspect of resonance amplitude A , which defines the modulation depth of the Fano lineshape, and strongly influences the change in scattered intensity for a given perturbation. A recent analysis by ACS Photonics 2022, 9, 1757–1763 has actually shown that if the resonance amplitude is taken into consideration, that the best sensing performance does not necessarily corresponds to the scenario of highest Q-factor, but instead corresponds to maximizing the product QA . Indeed, since the resonance amplitude of the meta ring increases versus increasing ΔL , as evident from Figure 3c, a similar balancing argument between Q-factor and amplitude can be applied to the meta rings in this work. Furthermore, the novelty of Fourier space analysis of resonant scattering requires a new performance metric that quantifies the degree of directional scattering induced by the perturbation, which would be an interesting topic for future follow up works. All these factors combined influence the ultimate Fisher information plotted in Figure 5 of the main text and are responsible for the saturation in Fisher information beyond $\Delta L = 80$ nm.

Modifications

We have extended the discussion starting on page 23, line 428 to better clarify concrete reasons for the saturation in Fisher information beyond $\Delta L = 80$ nm as follows:

“From the scalebar we can finally observe that the maximum information content increases with increasing asymmetry ΔL , ultimately saturating beyond $\Delta L = 80$ nm. To clarify this behavior, we recall that the Fisher information performance metric is not only influenced by conventional factors such as resonance wavelength shifts and Q-factor, but also by the resonant amplitude that defines the modulation depth of the Fano lineshape. Indeed, a recent study [47] has demonstrated that optimal sensing performance does not necessarily corresponds to the scenario of highest Q-factor, but rather to balancing the Q-factor and resonance amplitude. Since the resonance amplitude increases versus increasing ΔL , as evident from Figure 3c, a similar argument can be applied to the meta ring scattering sensor. Furthermore, the novelty of Fourier space analysis of resonant scattering requires a new performance metric that quantifies the degree of directional scattering induced by a perturbation, which would be an interesting topic for future investigations. All these factors combined influence the ultimate Fisher information plotted in Figure 5d and are responsible for the saturation in Fisher information beyond $\Delta L = 80$ nm.”

We have further added the corresponding reference 47:

[47] Conteduca et. al, “Beyond Q: the Importance of the Resonance Amplitude for Photonic Sensors”, ACS Photonics 2022, 9, 1757–1763

Reviewer comment

3. *The direct interpretation and discussion of how theoretical eigenmode changes in Figure 2 manifest as experimental Fourier space asymmetries in Figure 4 should be strengthened. Specifically, an explicit explanation linking the perturbation of the p_{dark} mode to the observed directional scattering is needed to enhance the connection between theory and experiment.*

Authors' response

The directional scattering patterns shown in Figure 2j and Figure 4 of the main text can be explained by considering the *full unperturbed and perturbed driven dipole configurations* \mathbf{p}_{driven}^0 and $\mathbf{p}_{driven}^{\Delta y}$, for which the scattered far-fields are directly calculated using the Green's function. Since a similar beaming effect is visible in the calculated and experimental differential Fourier space image of Figure 2j and Figure 4, a simple dipole model can already explain important physical features of the experiment. The driven dipole configurations can indeed be expressed as a sum of eigenmodes according to equation (1) of the main manuscript, effectively consisting of \mathbf{p}_{bright} and \mathbf{p}_{dark} . Crucially, the observed Fourier space radiation pattern will be *an interferogram* consisting of the radiation patterns emitted by each individual eigenmode, as is also visible from equation (4). All modes, with corresponding (perturbed) expansion coefficients, should therefore be taken into account to explain the observed asymmetry in Fourier space. This interferometric character fundamentally underlies the high sensitivity.

Modifications

To better connect the perturbations of individual eigenmodes to the observed directional scattering events in Fourier space, we have rephrased the following sentence on page 11, line 201:

“Perturbations imprinted on the individual eigenmodes would in turn lead to corresponding changes in the modal radiation patterns and, through equation (4), on the total radiation pattern $|S_{tot}(\hat{\mathbf{k}}, \omega)|^2$ as well, which could lead to directional scattering events in Fourier space”

To highlight the connection between theory and experimental results, we added the following sentence on page 17, line 309:

“By comparing the calculated differential Fourier space image of Figure 2j with the images of Figure 4, we see that a simple dipole model can reproduce the salient features of the experiment.”

Reviewer comment

4. The apparent author's note "Radiation damping?" in the Methods Section should be removed or revised into a complete statement.

Authors' response

The “?” comment behind “radiation damping” in the Methods Section was due to an incorrect citation caused by a missing reference, we thank the reviewer for noticing this. We fixed this issue, as well as the “?” comment on line 505, by adding the correct references to the bibliography file.

Modifications

The citations are modified as follows:

Page 27, line 505: “<...> particles in the metarings as prolate spheroids⁴⁸, with a long axis <...>”

Page 28, line 509: “<...> Radiation damping⁴⁹ is included to the static polarizabilities <...>”

The corresponding references are added:

[48] Bohren et. al, “Absorption and scattering of light by small particles”, John Wiley & Sons, 2008

[49] Weber et. al, “Propagation of optical excitations by dipolar interactions in metal nanoparticle chains”, Phys. Rev. B., 2004, 70, 125429

Reviewer comment

5. A brief consideration of whether the proposed methodology is a general principle applicable to other types of Fano resonant systems or perturbations beyond the specific system studied would be welcome when discussing the general applicability of Fourier space analysis.

Authors' response

We thank the reviewer for mentioning this important point. The crucial message of the paper is that angle resolved analysis of resonant scattering is *generally* more informative than angle integrated analysis. A Fano resonance is traditionally viewed as a near field coupling effect between a dark and bright eigenmode. A hitherto unrecognized fact is that the optical theorem implies that near-field Fano effects necessarily imply that any Fano resonance manifests itself as modal interferogram of the radiation patterns of the respective modes. This provides an interferometric advantage for sensing, generating specific sensitivities as a function of viewing angle that would otherwise be obscured in a conventional spectroscopic measurement. The necessary condition is that the Fano resonant structure is able to scatter light into higher order Fourier channels. To support the generality of the approach, we have furthermore performed theoretical calculations in which we compare the information content between Fourier space images and scattering spectra for different Fano resonant structures and perturbation types beyond those investigated in the main text.

Modifications

To highlight the generality of the work, we have added the following paragraph on page 24, line 443:

“We finally note that the information advantage of a Fourier space measurement with respect to an angle integrated measurement is generally valid for other Fano-resonant structures and perturbation types beyond those investigated in this paper. The crucial necessity is that the photonic structure should be able to scatter light into higher-order diffraction channels. The generality of the approach can be understood by viewing the Fano-resonance not only as a near-field coupling effect between bright and dark modes: the optical theorem fundamentally implies that any near-field effect that gives rise to a Fano line also expresses as a far-field modal interferogram. This interferometric advantage can generate specific sensitivities as a function of viewing angle due to perturbations. To support this claim, we have performed coupled dipole calculations on different Fano-resonant structures and perturbation types in the SI, all of which demonstrate the information advantage of Fourier space readout.”

Likewise, we have added a new section “Different Fano structures and perturbation types” to the supplementary information, in which we show and discuss results of coupled dipole calculations on different Fano-resonant structures and perturbation types.

Reviewer comment

6. Consider adding brief explanations for terms like “interlaced metrology” that might be unfamiliar to readers outside that specific field to add a terminology explanation.

Authors' response

We agree that more elaboration on “interlaced metrology” is required, and have added a brief description in the text to introduce the specific terminology.

Modifications

We added the following description on page 6, line 107:

“In the following, we will focus on sensing perturbations in the form of deeply subwavelength structural displacements between the individual building blocks in meta rings as described above. This type of perturbation is frequently encountered in modern-day semiconductor chip manufacturing processes, where nanometer-scale misalignments between individual structures on a device layer can arise when the same wafer goes through several successive exposure/etch/exposure cycles where each exposure step only defines part of the structure. Accurate monitoring of these misalignments, which is termed “interlaced metrology” [35,36], is crucial to ensure high-quality chip performance. This is typically achieved by analyzing the diffracted signal of devoted scattering sensors that are printed during the same exposure as the actual devices in the chip. Currently, there is a large push within the semiconductor industry for small footprint scattering sensors containing an effective footprint of no more than $2 \times 2 \mu\text{m}^2$. Since the meta rings contain a diameter of only $3 \mu\text{m}$, which could be reduced even further if needed, we envision that these structures could be exploited as responsive, yet small footprint nanophotonic scattering sensors for diffraction based interlaced metrology applications.”

Reviewer comment

7. The logical connection between the cancellation by the interference term and the similarity of modal radiation patterns should be clarified to reinforce the logic in Equation (4) discussion.

Authors' response

The crucial connection here is the optical theorem, in combination with the conservation of photon flux. Assuming that there is *complete* Fano transparency in extinction at a certain frequency and that there is no absorption, that would result in $\sigma_{scat} = 0$. Since σ_{scat} integrates the total radiation pattern over all viewing angles as $\sigma_{scat} \propto \int |S_{tot}(\hat{\mathbf{k}}, \omega)|^2 d\hat{\mathbf{k}}$, that would mean that $|S_{tot}(\hat{\mathbf{k}}, \omega)|^2 = 0$ for every viewing angle. Assuming that \mathbf{p}_{bright} and \mathbf{p}_{dark} are the only modes of the system, from equation (4) it can then be deduced that $S_{bright}(\hat{\mathbf{k}})$ and $S_{dark}(\hat{\mathbf{k}})$ should be identical in shape, and that the expansion coefficients $\alpha_{bright}(\omega)$ and $\alpha_{dark}(\omega)$ tune the relative transparency.

Modifications

We extended the logic around equation (4) and the similarity between modal radiation patterns for further clarification on page 11, line 195:

“Suppose there is complete transparency in extinction at a certain frequency and that there is no absorption. In that case, the optical theorem states that $\sigma_{scat} = 0$. Since $\sigma_{scat} \propto \int |S_{tot}(\hat{\mathbf{k}}, \omega)|^2 d\hat{\mathbf{k}}$, that would mean that the interferometric term in equation (4) fully cancels the first two terms for every $\hat{\mathbf{k}}$. By conservation of photon flux it can then be concluded that the modal radiation patterns $S_{bright}(\hat{\mathbf{k}})$ and $S_{dark}(\hat{\mathbf{k}})$ should be very similar to each other, and that the expansion coefficients $\alpha_{bright}(\omega)$ and $\alpha_{dark}(\omega)$ tune the relative transparency.

Reviewer comment

8. Enhance the readability of the optical component list by adjusting text size or explicit mention in the caption to improve the readability of Figure S1 in Supplementary Information.

Modifications

We increased the font size of the optical components list in Figure S1 for better readability.

Reviewer comment

9. A sentence or two commenting on the validity of the Gaussian noise assumption for Fisher information calculation would be appropriate when briefly discussing the Gaussian noise assumption.

Authors' response

We thank the referee for raising this point. As noted in the manuscript (page 19, line 361), we justify the Gaussian noise assumption by explaining that a Gaussian conveniently models the contributions from mixed noise sources, while Poisson distributed noise approximates Gaussian noise for the count rates relevant in our experiment. To avoid any ambiguity, we have slightly rephrased this passage to explicitly state that this makes the Gaussian noise model valid for the Fisher information calculation in our regime.

Modifications

We rephrased the justification for Gaussian noise on page 19, line 361 as follows:

“The choice of a Gaussian distribution instead of Poissonian is a convenient way to account for the mixed contributions from shot noise and detector readout noise, and for the count rates at hand the Gaussian distribution approaches the Poissonian one. Using this Gaussian noise model we can therefore make reasonable estimations of the available Fisher information in the data by using a closed form analytical expression, which is obtained by substituting equations (7) and (8) into equation (5).”

We added the following sentence on page 20 line 371, to highlight different approaches to estimate the Fisher information beyond closed form analytical expressions.

“Besides analytics, we note that model-free approaches exist to determine the available Fisher information from data where the noise model is inherently unknown [45]”

[45] Starshynov et. al, “Model-free estimation of the Cramer–Rao bound for deep learning microscopy in complex media”, Nat. Photonics 2025, 19, 593–600

Reviewer 3

Reviewer comment

This manuscript presents a study demonstrating that angle-resolved measurements of light scattered from Fano-resonant dielectric metasurfaces provide significantly more information than conventional spectroscopic (angle-integrated) detection strategies. The authors quantify this improvement using Fisher information theory and validate the concept theoretically and experimentally. I find this work to be well executed in both its theoretical and experimental parts, and exploiting the angular (Fourier-space) information content of the far field appears to be a novel concept in resonant nanophotonic sensing, going beyond typical sensing schemes involving only the spectral resonance shift. The finding of a sevenfold increase in information demonstrates the advantage of this approach (under idealized conditions).

Authors' response

We thank reviewer 3 for their kind comments and detailed feedback, in the following we will address all suggestions point by point.

Reviewer comment

In the following section, I list a few points for the authors to consider:

- 1. What are the prospects of generalising this approach to other common Fano-resonant structures?*

Authors' response

We thank the reviewer for mentioning this important point. The crucial message of the paper is that angle resolved analysis of resonant scattering is *generally* more informative than angle integrated analysis. A Fano resonance is traditionally viewed as a near field coupling effect between a dark and bright eigenmode. When measuring *observables* in the far-field, however, the Fano resonance manifests itself as modal interferogram of the radiation patterns of the respective modes. This interferogram can generate specific sensitivities as a function of viewing angle that would otherwise be obscured in a conventional spectroscopic measurement. The important necessity is that the Fano resonant structure is able to scatter light into scattering angles, and not just the specular direction. To support the generality of the approach, we have performed theoretical calculations in which we compare the information content between Fourier space images and scattering spectra for different Fano resonant structures and perturbation types beyond those investigated in the main text.

Modifications

To highlight the generality of the work, we have added the following paragraph on page 24, line 443:

“We finally note that the information advantage of a Fourier space measurement with respect to an angle integrated measurement is generally valid for other Fano-resonant structures and perturbation types beyond those investigated in this paper. The crucial necessity is that the photonic structure should be able to scatter light into higher-order diffraction channels. The generality of the approach can be understood by viewing the Fano-resonance not only as a near-field coupling effect between bright and dark modes: the optical theorem fundamentally implies that any near-field effect that gives rise to a Fano line also expresses as a far-field modal interferogram. This interferometric advantage can generate specific sensitivities as a function of viewing angle due to perturbations. To

support this claim, we have performed coupled dipole calculations on different Fano-resonant structures and perturbation types in the SI, all of which demonstrate the information advantage of Fourier space readout.”

Likewise, we have added a new section “Different Fano structures and perturbation types” to the supplementary information, in which we show results of coupled dipole calculations on different Fano-resonant structures and perturbation types.

Reviewer comment

2. Both the perturbation and the excitation polarisation are aligned along the y-axis. How sensitive are the results with respect to this alignment? Would using different polarizations affect the directional sensitivity?

Authors' response

Because the specific perturbation Δy breaks the symmetry of the meta ring, we expect that different excitation polarization states would indeed influence the directional scattering effects in the meta ring as well. From an eigenmode perspective, Δy would lift the degeneracy between \mathbf{p}_{bright} and \mathbf{p}_{dark} along the x and y direction, resulting in different modal perturbations along x and y directions. These modal perturbations would in turn lead to different directional scattering fingerprints in Fourier space. Note that this only holds for perturbations that introduce asymmetry to the structure. In the Figure 1 below we show exploratory calculations of differential Fourier space images for different incident polarization states, showing the sensitivity of the directional scattering to the incident polarization state. An interesting question is which polarization state is most optimal for sensing perturbations Δy . More generally, supposing one could arbitrarily shape the incident excitation in polarization, amplitude and phase distributions (beyond a single channel), what would the optimally informative excitation be?

Figure 1: Sensitivity of directional scattering for different incident polarizations. (A): Calculated differential Fourier space image for a polarization state parallel to the perturbation. (B): Calculated differential Fourier space image for a polarization state perpendicular to the perturbation. Both Fourier images are calculated at a wavelength of 775nm, with modeling parameters discussed in the main manuscript.

Modifications

To clarify the sensitivity of the directional scattering to the incident polarization, we have added the following sentences on page 17, line 313:

“It is important to note that the directional scattering effects are sensitive to the polarization state of the incident excitation, since the specific perturbation Δy breaks the degeneracy of the bright and dark modes along the x and y direction. An interesting question for future works would be to probe which (polarization) excitation scheme generates maximum information about a given perturbation, which could be tackled with the recently introduced framework of maximum information states [39].”

Reviewer comment

3. Can the authors provide a prospect on the practical applications of their work? Could it be useful in semiconductor processing or biosensing with fewer photons, for example?

Authors' response

We envision two practical application scenarios of this paper. First of all, the *specific* resonant meta rings could be used as highly sensitive scattering sensors to sense interlaced displacements within a device layer in next generation semiconductor optical metrology applications. Here, the meta rings provide a *resonant* response that is not readily available in state of the art metrology sensors composed of simple diffraction gratings. Furthermore, there is a large push in the semiconductor industry for small footprint scattering sensors containing a footprint of no more than $2 \times 2 \mu\text{m}^2$, as opposed to the $10 \times 10 \mu\text{m}^2$ targets that are currently used. The meta rings contain a diameter of only $3 \mu\text{m}$ that could be reduced even further if necessary, and thereby hold promise as highly responsive, yet small footprint sensors.

The second application connects to the earlier comment of the reviewer to the generality of the approach towards other Fano resonant structures and other perturbation types. Since the Fourier analysis of resonant scattering is *generically* more informative than conventional spectral analysis, this approach could also be used in different sensing contexts such as refractive index, particle or molecule sensing. Since Fisher information (in the shot-noise limit) scales with the number of detected photons, *a similar information content* can be retrieved from a Fourier space measurement compared to the conventional spectral approach, *for a lower photon budget*. This means that the integration time, or incident power could be reduced in a Fourier space experiment, while not losing Fisher information with respect to conventional spectroscopy.

Modifications

To better clarify the practical relevance of the work in the domain of semiconductor metrology, we have extended the paragraph on the application scenario of interlaced metrology on page 6, line 107:

“In the following, we will focus on sensing perturbations in the form of deeply subwavelength structural displacements between the individual building blocks in meta rings as described above. This type of perturbation is frequently encountered in modern-day semiconductor chip manufacturing processes, where nanometer-scale misalignments between individual structures on a device layer can arise when the same wafer goes through several successive exposure/etch/exposure cycles where each exposure step only defines part of the structure. Accurate monitoring of these misalignments, which is termed “interlaced metrology” [35,36] is crucial to ensure high-quality chip

performance. This is typically achieved by analyzing the diffracted signal of devoted scattering sensors that are printed during the same exposure as the actual devices in the chip. Currently, there is a large push within the semiconductor industry for small footprint scattering sensors containing an effective footprint of no more than $2 \times 2 \mu\text{m}^2$. Since the meta rings contain a diameter of only $3 \mu\text{m}$, which could be reduced even further if needed, we envision that these structures could be exploited as responsive, yet small footprint nanophotonic scattering sensors for diffraction based interlaced metrology applications.”

To elucidate the general implications of Fano resonant Fourier scatterometry in sensing experiments, we have included the following sentences on page 24, line 454:

“Since Fisher information (in shot-noise limited experiments) scales with the number of detected photons, a similar information content can be retrieved from a Fourier space measurement compared to conventional spectroscopy, yet for a lower photon budget. This means that the integration time, or incident power could be reduced in a Fourier space read out scheme, while not losing Fisher information with respect to conventional spectroscopy.”

Reviewer comment

4. Although this probably goes beyond the scope of this work, it could also be interesting to see if an artificial neural network can reach the detection limit resulting from the Fisher information analysis in this work (along the lines of Nature Photonics 19, 593–600 (2025)).

Authors' response

While this indeed goes beyond the scope of this paper, using an ANN could indeed be an interesting strategy to perform actual estimations of Δy from the Fourier scatterometry data, and we thank the reviewer for pointing this out. The Fisher information analysis performed in this work could then be used as a reference to assess the ultimate performance limit of such neural-network-based detection strategies.

Modifications

We have added the following sentence on page 26 line 482, along with the corresponding reference:

“Finally, estimators that reach the Cramer-Rao bound can be constructed to perform actual estimations of Δy by analysis of the Fourier scatterometry images. For this, artificial neural networks have been shown to efficiently reach the detection limit set by the Fisher information as determined in this work [45].”

[45] Starshynov et. al, “Model-free estimation of the Cramer–Rao bound for deep learning microscopy in complex media”, Nat. Photonics 2025, 19, 593–600

General reviewer comments

Comment reviewer 1

The authors made a very serious revision and addressed clearly my major concerns. I did like that they extended the study by adding other types of Fano-resonant structures and perturbations that are frequently employed by the sensing community. This new section "Different Fano structures and perturbation types" added to the supplementary part is very useful, and it cover different scenarios of Fano resonant structures and perturbation types.

I am happy to suggest publication in the current form

Comment reviewer 2

As a reviewer specializing in devices utilizing Fano resonances, I would like to clarify that my expertise lies primarily in the development and characterization of such photonic structures, rather than in the field of sensing methodologies or information theory. While I appreciate the authors' efforts to expand the sensing paradigm through the analysis of Fisher information in the Fourier space, I am not in a position to critically evaluate the novelty or impact of this approach from a sensing expert's perspective.

Comment reviewer 3

This manuscript NCOMMS-25-26949A has been seen by altogether three reviewers, who provided comments and suggestions.

From my point of view, the authors have carried out a thorough revision of their manuscript and have also provided a very detailed rebuttal letter in which all concerns by the referees have been addressed. Apart from a number of technical clarifications, the authors have also addressed the concern of Referee 1 regarding novelty: I strongly agree here with the authors, that the information stored in a Fourier space analysis clearly goes beyond an analysis of the spectral lineshape only.

In light of the above, I strongly recommend this manuscript for publication in Nature Communications.

Authors' response

We would like to thank all reviewers for their time and efforts to review our work. We feel that the message of our paper has improved substantially thanks to the critical comments of the reviewers, and are glad that all 3 reviewers were unanimously impressed with the quality of the manuscript and corresponding revisions.

Minor revisions reviewer 3

Reviewer comment

Minor points to clarify at the proof stage:

1. In line 434 there is a typo "...does not necessarily corresponds..."

Modifications

We have corrected the typo in line 434.

Reviewer comment

2. Typically the definition of the S -matrix does not only involve the scattered part, but also the incident part (in contrast to the transition matrix T). In this sense, the scattering cross section depends only on the scattered field, not on the incident one (see expressions below Eq. (4)). I assume this is just an issue of convention though, where the authors denote S_{tot} to be the matrix that describes "total radiated (scattered) field from all modes" rather than "total field = incident + scattered field", correct? But maybe I also just misunderstood something here.

Authors' response

Indeed, with S_{tot} we mean the total *scattered* field due to all eigenmodes, and does not include the incident fields.

Modifications

To avoid confusion, we have explicitly added the word "scattered" to the sentence "The acquired Fourier space *scattered* radiation diagrams..." before equation (3) on page 7.